# Measuring Quality of Life: Incorporating Objectively Measurable Parameters within the Cross-Sectional Bern Cohort Study 2014 (BeCS-14)

**DOI:** 10.3390/ijerph21010094

**Published:** 2024-01-15

**Authors:** Susanne Theis, Norman Bitterlich, Michael von Wolff, Petra Stute

**Affiliations:** 1Department of Obstetrics and Gynecology, University Medical Center of the Johannes Gutenberg University Mainz, Division of Gynecological Endocrinology and Reproductive Medicine, Langenbeckstr. 1, 55131 Mainz, Germany; susanne.theis2@unimedizin-mainz.de; 2Independent Researcher, Draisdorfer Str. 21, 09114 Chemnitz, Germany; norman.bitterlich@t-online.de; 3University Clinic Bern, Division of Gynecological Endocrinology and Reproductive Medicine, University Women’s Hospital, Inselspital Bern, Friedbühlstrasse 19, 3010 Bern, Switzerland; michael.vonwolff@insel.ch

**Keywords:** quality of life, subjective rating, objective rating, SF-36, biofunctional status, active and healthy aging

## Abstract

Up until now, the measurement of Quality of Life (QoL) was based on validated subjective rating tools rather than objective measurement. To become more independent of the self-assessment of probands, a way to objectively measure QoL should be found. A monocenter, cross-sectional, observational, non-interventional trial was performed from 2012 to 2014 at Inselspital Bern to evaluate the bio-functional status (BFS), a complex, generic, non-invasive, sex- and age-validated assessment tool, in a wide range of areas. A standardized battery of assessments was performed on 464 females and 166 males, ages 18 to 65 (n = 630). In addition to the survey of the BFS, participants replied—among others—to the validated questionnaire SF-36 for health-related QoL (n = 447, subgroup 1). Since the accepted cut-off value for BFA calculation is age ≥ 35 years, subgroup 2 included 227 subjects (all participants aged ≥ 35 years out of subgroup 1). In order to be able to compare the eight SF-36 subscales to BFS parameters, a comparable score set of single BFS items had to be constructed. Subsequently, we aimed to statistically identify BFS item combinations that best represented each SF-36 subscale. All eight SF-36 subscales were significantly represented by various different combinations of BFS items. A total of 24 single BFS items significantly correlated with SF-36 subscales, of which 15 were objective and nine were subjective. All eight SF-36 subscales were significantly represented by various different combinations of BFS items leading to stronger correlations (range five to nine BFS items), and overall, sex and age did not affect these associations, but in the SF-36 subscales ‘bodily pain’ (sex) and ‘role limitations due to physical health problems’ (age in men). To our knowledge, we are the first to correlate a validated set of 34 objective and 9 subjective parameters with subjectively evaluated SF-36 subscales. This first study on the objectifiability of the SF-36 questionnaire demonstrated that questions on quality of life can be answered independently of a subjective assessment by subjects in future scientific studies.

## 1. Introduction

Quality of Life (QoL) is a topic of broad interest, affecting all age groups. It is defined as “an overall general well-being that comprises objective descriptors and subjective evaluations of physical, material, social, and emotional well-being together with the extent of personal development and purposeful activity, all weighted by a personal set of values” [1]. In aging females and males, QoL is strongly affected by physical and mental health and, therefore, by Active and Healthy Aging (AHA). Accordingly, AHA incorporates biological aging, active aging, as well as changes in psychological and social well-being [2] that may support survival to old age, delay in the onset of chronic non-communicable diseases (NCD), and optimal functioning for the maximal time period at individual levels, body systems, and cells. The corresponding conceptual AHA framework [2] includes several items such as functioning (individual capability and underlying body systems), wellbeing, activities and participation, and diseases, including NCD. As such, AHA seems to be closely related to QoL [3]. There are various validated tools available for QoL assessment. However, these instruments have in common that they are based on subjective ratings rather than objective measurements. For this reason, we wanted to establish an independent way of measuring quality of life apart from subjective evaluation by a patient. At best, this could be used for future research on objective QoL.

The bio-functional status (BFS) assessment tool has been fitted into a theoretical model incorporating both the International Classification of Functioning (ICF) and AHA concepts. This complex, generic, non-invasive, sex- and age-validated assessment tool meets the European Innovation Partnership on Active and Healthy Aging (EIP—AHA) requirements for a diagnostic AHA instrument. Their requirements comprise applicability to health and disease across age stages (non-pediatric and non-geriatric lifetime), easy, partly self- and proxy administration, and accordance with the ICF of the World Health Organization (WHO). As QoL may be one major factor influencing AHA, we postulate that the complex generic AHA assessment diagnostic tool also reflects QoL in a more objective way than previous instruments. The BFS assessment tool covers physical, mental-cognitive, emotional, and social domains. Thus, our hypothesis was that the BFS reflected QoL, as assessed by the validated questionnaire SF-36. As previous research on the quality of life, and in particular, on health-related quality of life, has so far mainly used subjective methods, but objectifiable comparisons are equally important for scientific comparability, we wanted to examine whether it is even possible to depict the quality of life (which is per se subjective) with the help of objectifiable parameters.

## 2. Materials and Methods

### 2.1. Study Population

Between 15.05.2012 and 04.07.2014 464 German-speaking women and 166 men aged 18 to 65 were recruited at the University Women’s Hospital, Inselspital Bern, Switzerland. Recruitment was performed by the principal investigator (PS), this study nurse, and fourteen doctoral students of the medical school, University of Bern, via personal contact (patients, colleagues, family, friends) and online advertisement (internet, intranet Inselspital Bern, social media). Exclusion criteria were pregnancy, acute diseases (e.g., fever, acute pain syndrome), and illiteracy. This study protocol was approved by the Cantonal Ethics Committee Bern (Ref.-Nr. KEK-BE: 023112), and written informed consent was obtained from each participant.

### 2.2. Study Design

The study design of the original study has been described before [2,4,5,6,7]. Briefly, this was a monocenter, cross-sectional, observational, non-interventional trial. All participants (Total cohort (TC), n = 630) within BeCS-14 followed a standardized battery of assessments consisting of a personal and family history, bio-functional status (BFS), and derived from it, bio-functional age (BFA), as well as validated questionnaires for depression and anxiety (HADS) [8], health-related quality of life (SF-36) [9], and chronic stress (TICS) [4,6,10], respectively. Participants were asked to participate in further assessments addressing “nutrition” by AD-EVA [2,11] and PATEF [2,12], “employees’ health” by IMPULS [13], “stress” by heart rate variability [14], and “cognition” by the validated test battery IGD [2,15]. The assessments and subgroups relevant to this publication are further described in Section 2.3.

### 2.3. Assessment Procedures

#### 2.3.1. Personal and Family History

Briefly, we assessed age, social status (partnership, having children, satisfaction with relationship and sex life), lifestyle (alcohol, tobacco, sport, sleep), and job status (highest educational degree, current field of work, job position, working hours, monthly gross income, presenteeism, absenteeism). Personal and family history further comprised information about malignancy, cardiovascular disease, breathing disorder, abdominal and urogenital disease, metabolic disorder, skin and/or hair disease, neuromuscular and psychiatric disorders, as well as bone and joint disease.

#### 2.3.2. Health-Related Quality of Life (hrQoL): SF-36 [9,16,17,18]

The SF-36 is a standardized, validated self-report questionnaire (patient-reported outcome measure (PROM)). It comprises eight subscales: physical functioning (PF, 10 questions), bodily pain (BP, 2 questions), role limitations due to physical health problems (RP, 4 questions), role limitations due to personal or emotional problems (RE, 3 questions), mental health (MH, 5 questions), social functioning (SF, 2 questions), vitality (VT, 4 questions), and general health perceptions (GH, 5 questions). Likert scales and yes/no options are used to assess function and well-being. For scoring, each subscale is transformed into a scale from 0 to 100 by a standardized algorithm. Two component scores are derived from these subscales: a physical component score (PCS, out of the PF, RP, BP, and GH scales) and a mental health component score (MCS, out of the VT, SF, RE, and MH scales). The SF-36 includes a single item perceiving change in health status over the last year (“self-reported health transition,” question 2) and therefore consists of 36 questions in total. The latter is not included in the scoring process. Higher scores on all subscales represent better health and functioning; a mean score of 50 is the normative value for all scales. Internal consistency and reliability are good (Cronbach’s alpha 0.95 (PCS) and 0.93 (MCS)). Complete datasets for SF-36 were available for 447 subjects in the total cohort (=subgroup 1 (S1)).

#### 2.3.3. Bio-Functional Status, Bio-Functional Age

The BFS was assessed by a comprehensive test battery developed by Poethig et al. and reported by others [2,4,5,6,19,20,21]. It is used for functionally diagnostic measurement of an individual’s vitality. The test battery comprises holistic characteristics from physical, sensory, cognitive-mental, and emotional-social areas that fit into a complex theoretical model incorporating the ICF and AHA concepts. The BFS test battery comprises, among others, 43 parameters that are used for BFA calculation. An overview of BFS assessment items sorted by subdomains of BFS (Appendix A) and a presentation of individual proceedings of BFS are shown in Appendix A. The test battery for BFS assessment is a validated age- and sex-specific tool (objectivity 0.96, reliability 0.93, female age validity: total age correlation 85.2%; total age commonality in the main factor 76.3%). The BFA (also called BFA-index) is based on a sex-specific regression and factor analysis of functional age [19,20,21,22]. For BFA calculation, the age of ≥ 35 years is the model-required accepted cut-off value (this results in subgroup 2 (S2) consisting of 227 subjects). The BFA-Index is the difference between chronological age (CA) and BFA (∆CA-BFA [year-equivalents]) and is an important characteristic of BFS.

### 2.4. Statistical Methods

The statistical analysis was performed with SPSS (Statistical Package for the Social Sciences) version 28.0.1.1 (14). The data have been analyzed descriptively. The descriptive statistics contained the calculation of the median, 1st and 3rd quartiles, mean, and standard deviation. As BFS data as well as SF-36 data were not normally distributed, non-parametric analyses were performed.

In order to be able to compare the eight SF-36 subscales (0–100 scale each) to BFS parameters, a comparable score set of yet-identified single BFS items had to be constructed. Significant non-parametric bivariate correlations between SF-36 subscales and BFS parameters were identified in the first step. A *p*–value < 0.05 was considered statistically significant. We used the 5%- and 95% percentile as lower and upper limits and then linearly transformed to 0 to 100 for each BFS parameter for score-building. Values above or below these limits were converted to zero or 100. The identified significant bivariate correlation BFS parameters were used to build an average.

Therefore, various BFS parameter combinations were tested to best represent each SF-36 subscale, and the combination with the best significant correlation coefficients was used. For this purpose, the method of backward elimination was applied. Sequentially excluding the primarily included variables was carried out. Step by step, each parameter for which the reduced mean value resulted in the largest correlation coefficient was removed. This procedure was terminated if the removal of a further parameter reduced the correlation coefficient. In cases of missing data, the mean value of available data were used as a substitute. This was tolerated for up to 25% of the parameters per subscale of a subject.

## 3. Results

### 3.1. Characteristics of the Cohort

The whole BeCS-14 cohort comprised 630 participants (total cohort (TC)). For SF-36, complete data sets were available for 447 subjects (subgroup 1 (S1)). A total of 227 participants had complete data sets for both SF-36 and BFS/BFA (subgroup 2 (S2)). Subgroup 1 was assumed to be representative of the total cohort, as their percentages and mean values fit into the 95% confidence interval of the total cohort. As BFA can only be calculated for individuals aged ≥ 35 years, the characteristics of subgroup 2 differed from those of the total cohort and subgroup 1 (Table 1 and Table 2). Briefly, the majority of the total cohort was female (73.7%), the mean age was 39.5 (SD 14.9%) years, 55.6% were married or lived in a permanent partnership, 64.9% were childless, and more than 50% were well-educated and had a monthly gross income above 5000 CHF. More than 50% worked in the social or economic sector, with 51.7% being employed. Job occupation was full-time with at least 90% capacity utilization in 41.6% and part-time from 50% to less than 90% capacity utilization in 26.7% of participants. The majority were non-smoking (65.0%) and physically active (74.6%). Approximately 35.2% consume alcohol daily. While only 24.1% were disease-free, regular pharmacotherapy was reported by 57.1% of participants. In the total cohort, there were 11 subjects with a personal history of a life-threatening cardiovascular event (seven subjects in S1 and S2).

### 3.2. Health-Related QUALITY of Life (SF-36)

Table 3 presents the eight SF-36 subscales (PF, BP, RP, RE, MH, SF, VT, and GH) of subgroup 1 (S1) (n = 447) in comparison to the SF-36 US validation cohort (n = 2474) [23,24]. Briefly, in S1, mean values in each of the eight SF-36 subscales were higher than in the US general population (which served as a reference for the validity of the SF-36), indicating a higher hr-QoL in our cohort. However, median values were similar for subscales RE, RP, SF, MH, and VT.

### 3.3. Bio-Functional Status (BFS) and Bio-Functional Age (BFA)

Appendix A presents descriptive BFS results of the total cohort (TC, n = 630), subgroup 1 (S1, n = 447), and subgroup 2 (S2, n = 227), as well as further determined parameters of the assessment process describing the cohorts but not used for BFA Index calculation. Briefly, mean CA was higher in S2 (52.3 ± 8.2 years) than in TC and S1 (each 39.6 ± 14.8 years), respectively. As BFA could only be calculated for individuals ≥ 35 years, mean BFA was comparable for TC (age > 35 years in TC n = 321): 44.0 ± 8.7 year-equivalents, S1 and S2: each 43.9 ± 8.7 year-equivalents. Accordingly, mean ∆CA-BFA was similar for S1 and S2 (8.6 ± 8.3 year-equivalents), as well as slightly higher than in TC (8.3 ± 8.1 year-equivalents), indicating that the TC was functionally slightly younger than S1 and S2, respectively. In the physical and sensory physiology BFS subdomains, chronologically older subjects (S2) performed worse than their younger counterparts (TC, S1) in some aspects, e.g., blood pressure, vital capacity, body composition (% fat mass), teeth status, athletic endurance (PPI), hand grip strength, vision, and hearing ability. Similarly, in the cognitive-mental BFS subdomain, chronologically older subjects (S2) performed worse in most BFS items, particularly in orientation and cognitive switching capability as well as in strategic thinking. In the emotional-social BFS subdomain, most BFS items were comparable between TC, S1, and S2. However, physical well-being was worse in the older subjects (S2).

### 3.4. Incorporating SF-36 into the Active and Healthy Aging Concept Represented by BFS

As described above (Section 2.4), we aimed to statistically identify BFS item combinations that best represented each SF-36 subscale to enable the most objective measurement of QoL possible. Table 4 presents the respective significant correlation coefficients for each SF-36 subscale, showing comparable correlation coefficients for TC, S1, and S2. Thus, in the following sections, we present the results for the TC.

Overall, there were 21 out of 43 single BFS items that significantly correlated with SF-36 subscales, some of them multiple (up to six) times (Table 5). Out of these 21 significantly correlating items, 13 were objective items, and eight of them were subjective items. Objective items that occurred several times were ranked between one (resting heart rate, pulse performance index (PPI), vital capacity, psychomotor endurance, tapping basic rate, visuomotor coordination ability, ability to concentrate, strategic thinking) and four times (fat mass, DMF). Subjective items that occurred several times ranked between one (social dominance) and six times (physical well-being, emotional well-being) occurrence of correlation with a SF-36 subscale.

#### 3.4.1. SF-36 Subscale General Health Perception (GH)

The best significant correlation coefficient (cc = 0.468) comprised six BFS items, covering three out of four BFS subdomains (physical, cognitive-mental, and emotional-social): systolic blood pressure, decayed missing filled teeth (DMF), optical reaction time, physical well-being, emotional well-being, and sense of coherence (SOC_L_9). The regression model for GH was statistically significant, both overall (F = 91.004, regression coefficient, rc = 0.472, *p* < 0.001) and for men (F = 21.036, rc = 0.544, *p* < 0.001) and women (F = 75.028, rc = 0.467, *p* < 0.001). Age did not affect the regression statistically significantly for men (*p* = 0.816) or for women (*p* = 0.120).

#### 3.4.2. SF-36 Subscale Role Limitations Due to Personal or Emotional Problems (RE)

The best significant correlation coefficient (cc = 0.363) comprised seven BFS items, covering three out of four BFS subdomains (physical, sensory physiology, and emotional-social): fat mass, psychomotor endurance, emotional well-being, physical well-being, sense of coherence (SOC_L_9), social dominance, and social power. The regression model for RE was statistically significant, both overall (F= 51.219, rc = 0.654, *p* < 0.001) and for men (F = 9.425, rc = 0.671, *p* = 0.003) and women (F = 43.869, rc = 0.670, *p* < 0.001). Age did not affect the regression statistically significantly for men (*p* = 0.509) or for women (*p* = 0.163).

#### 3.4.3. SF-36 Subscale Physical Functioning (PF)

The best significant correlation coefficient (cc = 0.427) comprised eight BFS items, covering three out of four BFS subdomains (physical, sensory-physiology, and emotional-social): pulse performance index (PPI), resting pulse, vital capacity DMF, active cell mass, visuomotor coordination ability, physical well-being, and sense of coherence (SOC_L_9). The regression model for PF was statistically significant, both overall (F = 47.300, rc = 0.268, *p* < 0.001) and for men (F = 17.933, rc = 0.352, *p* < 0.001) and women (F = 40.984, rc = 0.356, *p* < 0.001). While there is no statistically significant influence of age in women (*p* = 0.148), the influence of age in men turns out to be statistically significant (*p* = 0.013).

#### 3.4.4. SF-36 Subscale Role Limitations Due to Physical Health Problems (RP)

The best significant correlation coefficient (cc = 0.345) comprised seven BFS items, covering three out of four BFS subdomains (physical, cognitive-mental, and emotional-social): fat mass, DMF, strategic thinking, physical well-being, emotional well-being, social activity/leisure, and social stress/resonance. The regression model for RP was statistically significant, both overall (F = 52.793, rc = 0.622, *p* < 0.001) and for men (F = 17.839, rc = 0.786, *p* < 0.001) and women (F = 35.831, rc = 0.577, *p* < 0.001). Age did not affect the regression statistically significantly for men (*p* = 0.528) or for women (*p* = 0.261).

#### 3.4.5. SF-36 Subscale Bodily Pain (BP)

The best significant correlation coefficient (cc = 0.377) comprised five BFS items, covering three out of four BFS subdomains (physical, sensory physiology, and emotional-social): fat mass, DMF, tapping basic rate, emotional well-being, and physical well-being. The regression model for BP was statistically significant, both overall (F = 112.380, rc = 0.542, *p* < 0.001) and for men (F = 19.568, rc = 0.509, *p* < 0.001) and women (F = 88.236, rc = 0.562, *p* < 0.001). While there is no statistically significant influence of age in women (rc = 0.009, *p* = 0.907), the influence of age in men turns out to be statistically significant (rc = -0.398, *p* < 0.001).

#### 3.4.6. SF-36 Subscale Social Functioning (SF)

The best significant correlation coefficient (cc = 0.464) comprised eight BFS items, covering three out of four BFS subdomains (physical, cognitive-mental, and emotional-social): fat mass, ability to concentrate, optical reaction time, physical well-being, emotional well-being, sense of coherence (Bio_Soc_L_9), social activity/leisure, and social power. The regression model for SF was statistically significant, both overall (F = 100.518, rc = 0.561, *p* < 0.001) and for men (F = 18.099, rc = 0.418, *p* < 0.001) and women (F = 81.079, rc = 0.611, *p* < 0.001). Age did not affect the regression statistically significantly for men (*p* = 0.482) or for women (*p* = 0.998).

#### 3.4.7. SF-36 Subscale Mental Health (MH)

The best significant correlation coefficient (cc = 0.547) comprised six BFS items, covering two out of four BFS subdomains (sensory physiology, emotional-social): active cell mass, physical well-being, emotional well-being, sense of coherence (Soc_L_9), social activity/duties, and social power. The regression model for MH was statistically significant, both overall (F = 100.518, rc = 0.561, *p* < 0.001) and for men (F = 18.009, rc = 0.418, *p* < 0.001) and women (F = 81.079, rc = 0.611, *p* < 0.001). Age did not affect the regression statistically significantly for men (*p* = 0.482) or for women (*p* = 0.998).

#### 3.4.8. SF-36 Subscale Vitality (VT)

The best significant correlation coefficient (cc = 0.529) comprised six BFS items, covering two out of four BFS subdomains (physical, emotional-social): systolic blood pressure, emotional well-being, physical well-being, sense of coherence (SOC_L_9), social activity/duties, and social stress/resonance. The regression model for VT was statistically significant, both overall (F= 137.475, rc = 0.425, *p* < 0.001) and for men (F = 43.528, rc = 0.527, *p* < 0.001) and women (F = 107.963, rc = 0.437, *p* < 0.001). Age did not affect the regression statistically significantly for men (*p* = 0.687) or for women (*p* = 0.637).

#### 3.4.9. SF-36 Single Item Perceiving Change in Health Status over the Last Year (Question 2)

There were no significant correlations between SF-36 question 2 and any BFS item.

## 4. Discussion

Health-related QoL is usually assessed by validated questionnaires, thus only incorporating an individual’s subjective ratings. Fortunately, as quality of life moves to the forefront of clinical research, its objectifiability and, above all, objective measurability become important for future research projects.

Based on our cross-sectional cohort study of BeCS-14 in mainly well-educated, middle-aged, healthy, and functionally age-delayed participants with good hr-QoL, we aimed to demonstrate that hr-QoL could also be assessed by objective parameters or a combination of subjective and objective parameters, respectively.

We found that (1) 24 single BFS items significantly correlated with SF-36 subscales, of which 15 objectives and nine subjectives items; (2) all eight SF-36 subscales were significantly represented by various different combinations of BFS items, leading to stronger correlations (range of five to nine BFS items); and (3) overall, sex and age did not affect these associations, but in the SF-36 subscales ‘bodily pain’ (sex) and ‘role limitations due to physical health problems’ (age in men).

To our knowledge, we are the first to correlate a validated set of 34 objectives and 9 subjectives parameters with subjectively evaluated SF-36 subscales. Out of these, 21 BFS items (13 objectives and 8 subjectives BFS items) were identified to significantly correlate at least once with the SF-36 subscales. In the following, we discuss the BFS item combinations that represented the respective SF-36 subscales and focus on the discussion of the objective parameters, since the correlation of subjective parameters of the SF-36 with further subjective measurements included in the BFS assessment is firstly expected and secondly not the focus of this work. Since it has been disputed so far whether the two sum scores of the SF-36 really represent the eight subscales reliably [25,26], we focused on the individual eight subscales in the evaluation as well as in the following discussion.

The SF-36 five-item subscale ‘general health’ (GH) measures overall self-rated health by assessing subjectively perceived general health, mobility, and the impact of physical and emotional limitations on everyday life [17]. The SF-36 subscale GH was represented by a combination of three objective and three subjective BFS items.

Blood pressure is known to be age-related [22]. Even without disease value, the aging process results in a loss of distensibility and compliance of arterial walls, resulting in lowered structural compliance of arterial walls and therefore an increase in blood pressure [27,28]. In addition, blood pressure as a measure of current physical performance reflects a lack of exercise [29] and stress [19]. As such, bp is a reliable objective parameter reflecting aging processes. Dental health status analyzed by the objective parameter DMF reflects oral health, which is an integral element of overall health and well-being. It is known that dental status decreases with aging [30]. Oral health enables eating, speaking, smiling, and socializing and may lead to discomfort, pain, and a reduced quality of life if restricted [31,32].

The third objective parameter, optical reaction time, represents vision as well as cuteness in terms of intelligence performance that is not dependent on education [22,33]. In addition, conclusions on cognitive processing as well as the following motor reactions can be drawn from this parameter [22]. Vision per se, known as age-dependent due to, e.g., reduction of accommodation, accommodation width, or increasing corneal or lens opacity, does not appear as a correlating factor of BFS with any SF-36 subscale in our study [34]. This is quite understandable since most limitations in this respect can be easily compensated by visual aids today.

Sense of coherence (SOC_L_9) as one of the subjective correlating BFS parameters is based on the concept of salutogenesis by Antonovsky, e.g., comprehensibility (the belief that things happen in a predictable fashion), anageability (the belief that the resources necessary to take care of things are available), and meaningfulness (the belief that things in life are a source of satisfaction) [35]. The original questionnaire developed by Antonovsky was adapted and validated in German [36,37]. The two remaining subjective parameters—physical and emotional well-being—reflect subjective assessments of subjects about their state of health and have been proven to be dependent on biological age. The underlying questionnaire is based on WHO criteria and distinguishes subjects with functional neurotic disorders from so-called normal mental subjects [22].

The SF-36 subscale “role limitations due to personal or emotional problems” (RE) assesses subjectively perceived difficulties in daily private and working life due to mental health problems (e.g., fear or despondency). It comprises the amount of time spent on work or other activities, the amount of work accomplished, and the care with which work is performed. The SF-36 subscale RE was best represented by a combination of two objective and four subjective parameters. The parameter fat mass objectively describes the nutritional situation of a subject. Fat mass can obviously have an impact on QoL of a subject. This fact has already been studied several times [38,39]. From a young age to about 75 years old, body weight and, consequently, BMI usually increase [40]. Psychomotor endurance, as part of the tapping test for checking the basic psychomotor speed and functional maturity, characterizes psychomotor behavior and agility [39,41]. The influence of psychomotor activity on quality of life has already been proven.

The “Giessen Test” examines various dimensions of social competence. One of them refers to authoritative role behavior (social dominance), and the second correlates to emotional adaptability (social power). It is already known from the literature that social competencies improve with increasing age due to longer experience in a professional or private environment [19]. Again, the subjective parameters of sense of coherence and physical well-being showed a correlation to the SF-36 subscale of RE. The subscore on subjectively perceived difficulties in daily private and working life due to mental health is thus understandably influenced by more subjective parameters. Nevertheless, objective factors that have an influence on this subscore could be identified.

The SF-36 subscale “role limitations due to physical functioning” (PF) assesses physical limitations in a range of activities, from vigorous exercise to performing self-care activities [42]. The SF-36 subscale PF was best represented by a combination of five objective and three subjective parameters. The three objective parameters—resting heart rate; pulse performance index; and vital capacity—reflect the cardiovascular function of a subject and thus his or her physical performance and resilience. The pulse performance index (PPI) is used to assess the training status of the heart and circulation, as well as the metabolism. It is an indicator of the stressability and performance of the system (cardio fitness). Values below one are to be classified as poor. We find them in people who have a lack of exercise or exercise incorrectly. Values above two are considered good. Optimally training endurance athletes can reach values of four [22]. Resting heart rate and vital capacity are two other objective physical functioning parameters that reflect heart rate regulation and lung function. These mentioned circle run parameter correlations to the subscale PF objectively reflect physical function, which the SF-36 subjectively queries. Knowing body size and height, with the help of bioimpedance measurement, capacitive resistance, and resistance can be determined. Out of these, the fat mass and the active cell mass of a body can be calculated. DMF, as described above, reflects aging demonstrably via oral status evaluation. Again, the subjective parameters of sense of coherence and physical well-being were part of the best correlation coefficient combination, showing the impact of these already introduced and explained parameters. Since physical health problems are described in subscale PF of SF-36, it seems comprehensible that three objective parameters of the BFS score concern physical performance, as well as the fact that six objective parameters reflect this subscale of SF-36. Since the statistical model, as described in 3.4.3, shows only scarce significance for men, it must be pointed out at this point that the aforementioned statements apply to women of all age groups. For men, the influences seem to diminish with increasing age. It must therefore be noted that our model for subscale PF cannot be safely applied to aging men. Since the reference group of men showed a low number of subjects, the correctness of the model should be verified on further subjects.

The SF-36 subscale “limitations due to physical health problems” (RP) assesses limitations in various roles, including work and daily activities, of a subject. This SF-36 subscale was represented by a combination of three objective and four subjective parameters.

Fat mass and DMF, as described above, are reproducible, age-dependent parameters. The third objective parameter, strategic thinking, as part of the labyrinth test indicates problem-solving behavior as a part of memory performance, as verified by this test. The underlying principle is based on the standardized evaluation of error and time values. It is known that all four dimensions examined in this test (orientation ability, planning ability, problem-solving behavior, and conversion ability) decline with increasing age. In particular, the ability to concentrate to perform this test procedure decreases with age [30]. Thus, it appears that the subscale RP of SF-36 can be represented by objectifiable parameters of memory performance.

Out of the four subjective parameters, two are repeatedly known and already explained in detail above the physical and emotional well-being score. Further subjective parameters that correlated with the SF-36 subscale RP covered stress exposure as part of the “Giessen Test” and social leisure/activity as part of the “Leningrader Erfassungsskala.” With the Giessen test, psychosocial dimensions of social competence are standardized but subjectively questioned. As a rule, social learning improves with age and can therefore be used as a parameter for assessing aging processes [19]. Stress exposition in this context means social well-being, activity, and integration. It is well known that social integration and participation in social life have a significant influence on aging as well as on QoL in old age [43]. Fitting to this realization is also that a correlation with social activity concerning leisure time (which more often includes social contacts) is shown. In summary, the subscale RP of SF-36 does not seem to be influenced by physical complaints alone but also by subjective parameters, which primarily concern social interaction and their own evaluation.

The SF-36 subscale “bodily pain” (BP) measures body pain intensity and the extent to which pain interferes with daily activities. This SF-36 subscale was represented by a combination of three objective and one subjective parameter. DMF, as one of the objective parameters described in detail above, has shown several times that oral health as well as tooth-related pain have an impact on aging as well as quality of life [32,44]. Fat mass and associated obesity as an increasing global health issue are well known to affect well-being, physical function, and mental function, reflecting the functional effects of a disease as perceived by the patient [45]. It is equally known that obesity and associated diseases (mostly non-communicable diseases) are associated with increased pain sensation [46]. Therefore, it is not surprising that the subjective feeling of physical well-being also has an impact on the subscale BP of SF-36. The third objective parameter, psychomotor endurance, characterizes psychomotor behavior and agility, as described above. Summarized, these results appear surprising according to current knowledge that pain perception can be worsened by mental well-being (for example, tested by SF-36) [47]. We can cautiously claim that older men are more receptive to bodily pain (in women, age has no influence) and that the correlation of BFS is overlaid by age.

The SF-36 subscale of Social Functioning (SF) assesses the impact of physical and mental health on social functioning. This SF-36 subscale was represented by a combination of three objective and five subjective parameters. The first objective parameter is the already-known parameter of fat mass. The other two objective parameters reflect the cognitive and mental performance of a subject. While optical reaction time assesses cuteness as described above, the ability to concentrate assesses concentration capacity, which is known to potentially decrease while aging and therefore is a common assessment in aging diagnostics [19,48]. Out of the five subjective parameters, social power, physical and emotional well-being, and sense of coherence correlated repeatedly. Social activity/leisure, as the fifth correlating parameter, describes the social ties and obligations of a subject. The fact that the largest share of this subscale belongs to the emotional and social areas is explainable by the fact that social functions are strongly dependent on these very factors as well as the fact that fat mass as an objectively measurable parameter and mental performance have an influence on the social function of a person.

The SF-36 subscale is Mental Health (MH): Assesses anxiety, depression, loss of behavioral/emotional control, and psychological well-being. This SF-36 subscale is represented by a combination of one objective and five subjective parameters. The distribution of parameters in this subscale of SF-36 reflects a strong influence of subjective appraisal (correlating parameters: social power, physical and emotional well-being, social activity/duty, sense of coherence) on a subject’s mental health and capacity. Nevertheless, the objective correlating parameter of active cell mass is a common factor in the prediction of the mental health of elderly people. It is known that low muscle mass is associated with poor cognitive function and quality of life, as well as higher depression symptoms in older patients [49]. Although the identified parameters are justifiable, a broader distribution of BFS parameters among the four subgroups would have been expected since the causes of mental illness are multifactorial [50].

The SF-36 subscale Vitality (VT) scale assesses vitality, energy level, and fatigue and is meant to be a measure of subjective well-being. This SF-36 subscale was represented by a combination of one objective and four subjective parameters. As described above, systolic blood pressure, the one objective parameter correlating to subscale vitality, is known for age-dependency. Surprisingly, all other correlating parameters (social stress, emotional well-being, social activity/ duty, sense of coherence) that have been described before in detail belong to the subjective area. Since vitality is likewise a multifactorial construct, a broader distribution of the parameters among the four subgroups of BFS parameters would have been expected [51]. Especially for the cognitive and mental subdomains of BFS, it is surprising that there is no connection to vitality. Nevertheless, this score had the strongest correlation coefficient.

The BFS and SF-36 are intended to objectively represent the quality of life at a specific point in time (not a time period). “SF-36 question 2” asks for the change in health status over the last year. Therefore, no correlation between BFS (asking for a point in time) and question 2, referring to the change in health status over the past year, is understandable.

Age and sex have no material impact on correlation coefficients in this investigation (except for the two subscales PF and BP of SF-36, where age had an impact on correlation in males and sex had an influence in the subscale BP of SF-36). Therefore, we postulate that our findings are a stable and objective option to measure QoL. Limited applicability for measuring quality of life in men might apply to the two subscales PF and BP. On the other hand, the proportion of men in the overall cohort is lower, so this could also be a cause for this result.

QoL is a complex concept, and within disciplines, it is interpreted and defined differently [52]. This already makes comparisons and investigations difficult. Furthermore, there are a variety of different existing instruments to measure QoL. Some of them are well-validated instruments after all. Nevertheless, different definitions as well as various measuring instruments have so far made scientific investigations and, in particular, the comparison of results extremely difficult. As far as we know, there are no validated instruments to measure QoL in an objective way. Therefore, our aim was to develop an attainable measurement tool for QoL in order to improve these difficulties in the future. We want to investigate, for example, the QoL of menopausal women at the workplace to achieve a long-term improvement for this group since we recently recognized that there are no satisfactory studies to date [53].

## 5. Conclusions

As described above, we demonstrated that hr-QoL could also be assessed by objective parameters or a combination of subjective and objective parameters. To our knowledge, we are the first to correlate a validated set of 34 objective and 9 subjective parameters with subjectively evaluated SF-36 subscales. The objectifiability of the SF-36 questionnaire demonstrates that questions on the quality of life can be answered independently of a subjective assessment by subjects in future scientific studies.

## Figures and Tables

**Table 1 ijerph-21-00094-t001:** Comparison of Total Cohort and Subgroups 1 and 2 (categorial values).

	Total Cohort (n = 630)		Subgroup 1 (n = 447)	Subgroup 2 (n = 227)
Parameter	%	95% CI	%	%
Male	26.3	22.7–29.9	27.3	20.7
Female	73.7	70.1–77.3	72.7	79.3
Single, living alone, widowed, divorced, or other	25.3	21.8–28.9	23.7	21.2
Living in a marriage	30.6	26.9–34.4	31.3	56.4
Childless	64.9	61.0–68.7	65.3	35.2
University degree	39.9	30.2–37.9	38.1	29.5
Advanced technical college	14.5	11.7–17.5	15.3	20.5
Vocational training	20.3	17.2–23.7	19.6	26.8
Monthly gross income < 5000 CHF	48.9	44.9–52.9	49.9	42.3
Monthly gross income > 5000 CHF	41.1	37.2–45.1	42.3	55.0
No own income	8.7	6.6–11.3	7.4	1.8
Professional field:				
Social	37.3	33.5–41.3	36.7	49.8
Economy	18.7	15.7–22.1	17.4	19.8
Jobless	3.0	1.8–4.7	3.4	3.5
Employee status:				
Executive position	17.1	14.2–20.3	17.0	26.0
Employed	51.7	47.7–55.8	52.1	67.4
Student	23.5	20.2–27.1	23.3	0.4
Job occupation				
>90%	41.6	37.7–45.6	43.4	39.2
50–89%	26.7	23.2–30.4	26.8	38.8
<50%	21.7	18.5–25.2	21.9	16.7
Non smoking	65.0	61.2–68.9	65.5	61.2
Physical activity untill sweating				
1–2 times/week	37.3	33.5–41.3	36.7	41.0
>2 times/week	37.3	33.5–41.3	38.0	28.2
Alcohol consumption up to 4 times/month	56.2	52.2–60.2	56.8	53.1
Alcohol consumption up to 4 times/week	6.2	4.4–8.4	6.7	7.5
Daily alcohol consumption of up to 3 glasses	35.2	31.5–39.2	34.9	37.0
Disease—free	24.1	20.8–27.7	24.6	25.1
No Medication use	42.9	38.9–46.9	42.7	33.1

Abbreviations: CI: Confidence interval of percentage; n: number; %: percentage; CHF: Swiss Franks.

**Table 2 ijerph-21-00094-t002:** Comparison of Total Cohorts and Subgroups (metric values).

	Total Cohort (n = 630)			Subgroup 1 (n = 447)		Subgroup 2 (n = 227)	
Parameter	Mean (SD) %	95% CI	Median Q1, Q3	Mean (SD) %	Median Q1, Q3	Mean (SD) %	Median Q1, Q3
Age	39.5	38.3–40.7	37	39.6	36	52.4	53
	(14.9)		25.0, 53.0	(14.9)	25.0, 53.0	(8.1)	47.0, 59.0
Children	1.76	1.6–1.9	1	1.8	1	2.5	3
	(1.17)		1.0, 3.0	(1.2)	1.0, 3.0	(1.3)	1.0, 3.0
Smoking (cigarettes/day)	1.62	1.53–1.7	1	1.6	1	1.58	1
	(1.06)		1.0, 2.0	(1.01)	1.0, 2.0	(0.91)	1.0, 2.0

Abbreviations: CI: Confidence interval of percentage; n: number; %: percentage; SD: standard deviation; Q1: Quartile 25%; Q3: Quartile 75%.

**Table 3 ijerph-21-00094-t003:** SF-36 subgroups for dataset with complete SF-36 data (subcohort 1) and U.S. general population.

	Subgroup 1 n= 477					U.S. General Population n = 2474	
SF-36 Subscore	Mean	SD	Min	Max	Median	Mean	Median
General health perceptions (GH)	77.5	16.0	20.0	100.0	80.0	72.0	72.0
Role limitations due to personal or emotional problems (RE)	88.7	25.6	0.0	100.0	100.0	81.3	100.0
Physical functioning (PF)	95.5	9.1	30.0	100.0	100.0	84.2	90.0
Role limitations due to physical health problems (RP)	90.1	23.8	0.0	100.0	100.0	81.0	100.0
Bodily Pain (BP)	83.2	21.3	0.0	100.0	84.00	75.2	74.0
Social functioning (SF)	88.3	17.1	0.0	100.0	100.0	83.3	100.0
Mental health (MH)	76.8	13.4	24.0	100.0	80.00	74.8	80.0
Vitality (VT)	63.0	16.1	15.0	100.0	65.0	60.9	65.0

Abbreviations: N: Number; SD: standard deviation; Min: Minimum; Max: Maximum.

**Table 4 ijerph-21-00094-t004:** Correlation coefficients BFS and SF-36.

	Total Cohort (n = 630)		Subgroup 1 (n = 447)		Subgroup 2 (n = 227)	
Parameter	CC	*p*	CC	*p*	CC	*p*
General health perceptions (GH)	0.468	<0.001	0.464	<0.001	0.476	<0.001
Role limitations due to personal or emotional problems (RE)	0.363	<0.001	0.372	<0.001	0.371	<0.001
Physical functioning (PF)	0.427	<0.001	0.430	<0.001	0.420	<0.001
Role limitations due to physical health problems (RP)	0.345	<0.001	0.346	<0.001	0.351	<0.001
Bodily Pain (BP)	0.377	<0.001	0.343	<0.001	0.329	<0.001
Social functioning (SF)	0.464	<0.001	0.468	<0.001	0.527	<0.001
Mental health (MH)	0.547	<0.001	0.568	<0.001	0.614	<0.001
Vitality (VT)	0.529	<0.001	0.537	<0.001	0.545	<0.001

Abbreviations: CC: correlation coefficient, *p*: *p*-value, n: number.

**Table 5 ijerph-21-00094-t005:** Single BFS items (x) included in correlation coefficients divided by SF-36 subscale and sorted by objective (light blue) or subjective items (light green).

	SF-36 Subscore							
BFS Item	GH	RE	PF	RP	BP	SF	MH	VT
Physical parameters								
Systolic blood pressure	x							x
Resting heart rate/ pulse			x					
Pulse performance index (PPI)			x					
Vital capacity			x					
Fat mass		x		x	x	x		
Active cell mass			x				x	
DMF	x		x	x	x			
Sensory physiology and psychomotor parameters								
Psychomotor endurance		x						
Tapping the basic rate					x			
Viseomotor coordination ability			x					
Cognitive and mental parameters								
Optical reaction time	x					x		
Ability to concentrate						x		
Strategic thinking				x				
Emotional-social parameters								
Social dominance		x						
Social stress/resonance				x				x
Social power		x				x	x	
Physical well-being	x	x	x	x	x	x	x	x
Emotional well-being	x	x	x	x	x	x	x	x
Social activity/leisure				x		x		
Social activitiy/duties							x	x
Sense of coherence (SOC_L_9	x	x	x			x	x	x

## Data Availability

Data are contained within the article and Appendix A.

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
