# Peer review of "Measuring Quality of Life: Incorporating Objectively Measurable Parameters within the Cross-Sectional Bern Cohort Study 2014 (BeCS-14)"

_ijerph, 2024, doi:10.3390/ijerph21010094_

Round 1
Reviewer 1 Report
Comments and Suggestions for Authors
The authors have completed a comparison of the bio-functional status (BFS) tool, which includes both objective and subjective measures, with the standard and widely-used SF-36 assessment of health, which includes only self-reported items. Although the authors report some (predictable) correlations between the BFS and the SF-36, they have not demonstrated any way in which the BFS (which is much more labor-intensive to collect, given the extensive in-person physical measurements required) performs better than the SF-36 as a predictor of, say, morbidity or mortality.
1. Phrases in the abstract that are unclear unclear: Since the BFS can be applied from the age of 35, this resulted in subgroup 2 (n = 227)…. a comparable score set 24 of single BFS was constructed
2. Background is insufficient. Authors have not explained why an objective measure of QoL is necessary or preferrable. Authors have not explained why the existing SF-36 is not sufficient.
3. This data process makes no sense: “We used the 5% and 95% percentile as lower and upper limits for each BFS parameter for score-building. Values above or below these limits were converted to zero or 100.” It would have made more sense to winsorize the data.
4. It would be helpful to compare subgroups to total population on variables that indicate selectivity of sample: education, occupation, income, etc. It is unclear why they are compared only on age, children, and smoking.
5. Statistical approach is limited. Factor analyses, regression models, or structural equation approaches could have been applied to better effect.
Comments on the Quality of English Language
Proof-reading would be beneficial.
Author Response
We thank the reviewers for their valuable comments and suggestions for improvement. We have revised the manuscript according to the reviewers' suggestion. All changes have been highlighted in the manuscript. Enclosed are our responses point by point to the reviewers' comments and suggestions:
Reviewer 1
The authors have completed a comparison of the bio-functional status (BFS) tool, which includes both objective and subjective measures, with the standard and widely-used SF-36 assessment of health, which includes only self-reported items. Although the authors report some (predictable) correlations between the BFS and the SF-36, they have not demonstrated any way in which the BFS (which is much more labor-intensive to collect, given the extensive in-person physical measurements required) performs better than the SF-36 as a predictor of, say, morbidity or mortality.
Author´s reply:
We thank the reviewer for this justified question concerning the relevance of what we did.
Previous research on quality of life is difficult to compare due to the fact that quality of life is interpreted and defined in different ways within and between disciplines. Consequently, many different instruments exist and are used. Since many of these have been developed on empirical considerations and not from a definition or conceptual models, there is a lack of conceptual clarity and this may result in a threat to validity of research. [1]
The aim of the study was not to show an improved prediction of morbidity or mortality by means of one or the other method. Rather, we wanted to examine whether it is even possible to depict quality of life (which is per se subjective) with the help of objectifiable parameters. As previous research on quality of life and in particular on health-related quality of life has so far mainly used subjective methods, but objectifiable comparisons are equally important for scientific comparability, our results show, that it may be possible to assess quality of life more independently of a patient's subjective assessment.
In order to emphasize the aim of the work more clearly, the following adjustments have been made to the manuscript:
Page 1 Headline Line 2-3: Measuring quality of life incorporating objectively measurable parameters within the cross-sectional Bern cohort study 2014 (BeCS-14)
Page 2 paragraph 1 line 59-61 Introduction: For this reason, we wanted to establish an independent way of measuring quality of life apart from subjective evaluation by a patient. At best, this could be used for future research on objective QoL.
…
Page 2 paragraph 2 line 75-79: As previous research on quality of life and in particular on health-related quality of life has so far mainly used subjective methods, but objectifiable comparisons are equally important for scientific comparability we wanted to examine whether it is even possible to depict quality of life (which is per se subjective) with the help of objectifiable parameters.
Page 6 paragraph 4 line 219 – 221) Results (3.4): As described above (section 2.4) we aimed to statistically identify BFS item combinations that best represented each SF-36 subscale to enable the most objective measurement of hrQoL possible.
Comment 1: Phrases in the abstract that are unclear: Since the BFS can be applied from the age of 35, this resulted in subgroup 2 (n = 227) … a comparable score set 24 of single BFS was constructed
Author´s reply:
The reviewer raises an important point regarding comprehensibility in the abstract. Adjustments were made as follows (page 1 paragraph 1 line 23-26): Since the model required accepted cut-off value for BFA calculation is age ≥ 35 years, subgroup 2 included 227 subjects (all participants aged ≥ 35 years out of subgroup 1). In order to be able to compare the eight SF-36 subscales to BFS parameters, a comparable score set of single BFS items had to be constructed.
Comment 2: Background is insufficient. Authors have not explained why an objective measure of QoL is necessary or preferrable. Authors have not explained why the existing SF-36 is not sufficient.
Author´s reply:
We covered this point in the authors’ answer to reviewers introductory comment “The authors have completed a comparison…” above as well as in the corresponding adaptations of the manuscript.
Comment 3: This data process makes no sense: “We used the 5% and 95% percentile as lower and upper limits for each BFS parameter for score-building. Values above or below these limits were converted to zero or 100.” It would have made more sense to winsorize the data.
Author´s reply:
We appreciate this comment and the justified criticism. Without an interim step (that was performed by linear transformation) it would be nonsensical with the substitute values 0 and 100.
The performed procedure is a variant of Winsoring. The transformation turned out to be necessary in order to treat all parameters comparably in the scoring system (because we do not want to introduce weighting coefficients). Adjustments have been made as follows (page 4 paragraph 2 line 155-157): We used the 5%- and 95% percentile as lower and upper limits and then linearly transformed to 0 to 100 for each BFS parameter for score-building. Values above or below these limits were converted to zero or 100.
Comment 4: It would be helpful to compare subgroups to total population on variables that indicate selectivity of sample: education, occupation, income, etc. It is unclear why they are compared only on age, children, and smoking.
Author´s reply:
The reviewer raises an important aspect and we agree with the fact that comparison of subgroups with total cohort in characteristic features is meaningful. Thus, we present a comparison of 20 categorial values to describe and compare the cohort and sub-cohorts in Table 1a (comparison of percentages). (Table 1 a Page 4 and page 5 line 185-186)
The comparison of the mentioned values age, children, and smoking in Table 1b results of the fact that these categories are presented in metric values (comparison of Mean/Median). (Table 1 b page 5 line 187-188)
Comment 5: Statistical approach is limited. Factor analyses, regression models, or structural equation approaches could have been applied to better effect.
Author´s reply:
As described in the text (e.g. last sentence in the abstract), this is about the basic possibility. Therefore, we limited ourselves (especially with regard to reference 19) to simple regression models based on the scales of the SF -36 and the additively formed scores of the BFS. The optimization of the modeling is reserved for further investigations.
Authors response to Questions for General Evaluation
|
2. Questions for General Evaluation |
Reviewer’s Evaluation |
Response and Revisions |
|
Does the introduction provide sufficient background and include all relevant references? |
Must be improved |
Please find the revised version and explanation above in the authors´ first answer to the first general statement of the reviewer |
|
Are all the cited references relevant to the research? |
Can be improved |
Please see authors’ answer Reviewer 2 Comment 8 |
|
Is the research design appropriate? |
Can be improved |
Please see authors’ answer Reviewer 1 Comment 5 |
|
Are the methods adequately described? |
Can be improved |
Please see highlighted changes in manuscript 1.0 Introduction/ 2.4 Statistical methos / 3.4 Incorporating SF-36… |
|
Are the results clearly presented? |
Can be improved |
Please see authors’ answer Reviewer Comment 4 as well as highlighted changes in manuscript |
|
Are the conclusions supported by the results? |
Can be improved |
Please see authors answer Reviewer 2 Comment 6 and 7 |
References
- Felce, D. and J. Perry, Quality of life: Its definition and measurement. Research in Developmental Disabilities, 1995. 16(1): p. 51-74.
- Kuh, D., et al., A life-course approach to healthy ageing: maintaining physical capability. Proc Nutr Soc, 2014. 73(2): p. 237-48.
- Stute, P., et al., Measuring Active and Healthy Ageing: Applying a generic interdisciplinary assessment model incorporating ICF. The journal of nutrition, health & aging, 2017. 21(9): p. 1002-1009.
- Stute, P., et al., Measuring chronic stress exposure incorporating the active and healthy ageing (AHA) concept within the cross-sectional Bern cohort study 2014 (BeCS-14). BioPsychoSocial Medicine, 2019. 13(1): p. 2.
- Stute, P., et al., Measuring cognitive performance in way that incorporates the concept of active and healthy ageing (AHA). Maturitas, 2019. 125: p. 27-32.
- Arifi, D., et al., Impact of chronic stress exposure on cognitive performance incorporating the active and healthy aging (AHA) concept within the cross-sectional Bern Cohort Study 2014 (BeCS-14). Archives of Gynecology and Obstetrics, 2022. 305(4): p. 1021-1032.
- Mathieu, L., et al., Illness perception in overweight and obesity and impact on bio-functional age. Maturitas, 2017. 103: p. 92-93.
- Hamilton, M., A RATING SCALE FOR DEPRESSION. Journal of Neurology, Neurosurgery & Psychiatry, 1960. 23(1): p. 56.
- Bullinger, M., I. Kirchberger, and J. Ware, Der deutsche SF-36 Health Survey Übersetzung und psychometrische Testung eines krankheitsübergreifenden Instruments zur Erfassung der gesundheitsbezogenen Lebensqualität. Zeitschrift für Gesundheitswissenschaften= Journal of public health, 1995. 3(1): p. 21-36.
- Schulz, P., W. Schlotz, and P. Becker, Trierer Inventar zum chronischen stress (TICS)[Trier inventory for chronic stress (TICS)]. 2004.
- Ardelt-Gattinger, E., et al., AD–EVA-KJ: Interdisziplinäres Testsystem zur Diagnostik und Evaluation bei Adipositas und anderen durch Ess-und Bewegungsverhalten beieinflussbaren Krankheiten im Kindes-und Jugendalter. Monatsschrift für Kinderheilkunde, 2009. 157(Suppl. 2): p. 206.
- Zenz, H., C. Bischoff, and V. Hrabal, Patiententheoriefragebogen:(PATEF). 1996: Hogrefe, Verlag für Psychologie.
- Molnar, M., C. Haiden, and B. Geißler-Gruber, IMPULS-Broschüre und IMPULS-Test. Betriebliche Analyse der Arbeitsbedingungen. 2012, AUVA, AK, ÖGB, WKÖ (Hg.) Wien.
- Camm, A.J., et al., Heart rate variability: standards of measurement, physiological interpretation and clinical use. Task Force of the European Society of Cardiology and the North American Society of Pacing and Electrophysiology. 1996.
- Kalbe, E., et al., Das Inventar zur Gedächtnisdiagnostik: Vorstellung der endgültigen Version. Aktuelle Neurologie, 2006. 33(S 1): p. P301.
- Morfeld, M. and M. Bullinger, Der SF36 Health Survey zur Erhebung und Dokumentation gesundheitsbezogener Lebensqualität. Physikalische Medizin Rehabilitationsmedizin Kurortmedizin - PHYSIK MED REHABIL KURORTMEDI, 2008. 18: p. 250-255.
- Ware Jr, J.E. and C.D. Sherbourne, The MOS 36-item short-form health survey (SF-36): I. Conceptual framework and item selection. Medical care, 1992: p. 473-483.
- O'Dea, I., M.S. Hunter, and S. Anjos, Life satisfaction and health-related quality of life (SF-36) of middle-aged men and women. Climacteric, 1999. 2(2): p. 131-40.
- Poethig, D., Experimentelle Entwicklung eines klinischen Diagnostikmodelles zur Objektivierung des biologischen Alters des Menschen : (e. nicht-invasives Komplexverfahren zur ambulanten gerontolog. Funktionsdiagnostik). 1984, Leipzig, Univ., Diss. B, 1984 (Nicht f.d. Austausch).
- Ries, W. and D. Pöthig, Chronological and biological age. Exp Gerontol, 1984. 19(3): p. 211-6.
- Dean, W., et al., Biological aging measurement: Clinical applications. 1988: Center for Bio-Gerontology.
- Meißner-Pöthig, D., Vitalität und ärztliche Intervention: Vitalitätsdiagnostik: Grundlagen, Angebote, Konsequenzen. 1996: Hippokrates-Verlag.
- Ware, J.E.J., et al. How to score and interpret single-item health status measures: a manual for users of the SF-8™ Health Survey. 2001.
- Ware, J., et al., SF36 Health Survey: Manual and Interpretation Guide. Lincoln, RI: Quality Metric, Inc, 1993, 1993. 30.
- Simon, G.E., et al., SF-36 Summary Scores: Are Physical and Mental Health Truly Distinct. Medical Care, 1998. 36(4): p. 567-572.
- Wilson, D., J. Parsons, and G. Tucker, The SF-36 summary scales: Problems and solutions. Sozial- und Präventivmedizin, 2000. 45(6): p. 239-246.
- Fluckiger, L., et al., Differential Effects of Aging on Heart Rate Variability and Blood Pressure Variability. The Journals of Gerontology: Series A, 1999. 54(5): p. B219-B224.
- Marchais, S.J., et al., Arterial Compliance and Blood Pressure. Drugs, 1993. 46(2): p. 82-87.
- Mariano, I., et al., BLOOD PRESSURE RESPONSES TO STRESS AFTER CHRONIC PHYSICAL EXERCISE: A SYSTEMATIC REVIEW WITH META-ANALYSIS. Journal of Hypertension, 2021. 39: p. e402.
- Poethig, D., Experimental development of a clinical diagnostic model objectifying bio-functional age (ing) of human being. 1984, Habilitation thesis (dt.). German: National Library Leipzig.
- Roca, J.A.R., et al., Oral Status of Elderly Patients In Long Stay Centers: A Systematic Review. 2020.
- Peres, M.A., et al., Oral diseases: a global public health challenge. The Lancet, 2019. 394(10194): p. 249-260.
- Tan, H., K.G. Peres, and M.A. Peres, Retention of Teeth and Oral Health–Related Quality of Life. Journal of Dental Research, 2016. 95(12): p. 1350-1357.
- Owsley, C., Vision and Aging. Annual Review of Vision Science, 2016. 2(1): p. 255-271.
- Swenor, B.K., et al., Aging With Vision Loss: A Framework for Assessing the Impact of Visual Impairment on Older Adults. The Gerontologist, 2019. 60(6): p. 989-995.
- Antonovsky, A., Unraveling the mystery of health: How people manage stress and stay well. 1987: Jossey-bass.
- Schumacher, J., et al., The Antonovsky Sense of Coherence Scale. Test statistical evaluation of a representative population sample and construction of a brief scale. Psychotherapie, Psychosomatik, Medizinische Psychologie, 2000. 50(12): p. 472-482.
- Singer, S. and E. Brähler, Die» Sense of Coherence Scale «: Testhandbuch zur deutschen Version. 2007: Vandenhoeck & Ruprecht.
- Blüher, M., Obesity: global epidemiology and pathogenesis. Nature Reviews Endocrinology, 2019. 15(5): p. 288-298.
- Mancuso, P. and B. Bouchard, The impact of aging on adipose function and adipokine synthesis. Frontiers in endocrinology, 2019. 10: p. 137.
- Ponti, F., et al., Aging and imaging assessment of body composition: from fat to facts. Frontiers in endocrinology, 2020. 10: p. 861.
- Pizarro-Mena, R., et al., Effects of a Structured Multicomponent Physical Exercise Intervention on Quality of Life and Biopsychosocial Health among Chilean Older Adults from the Community with Controlled Multimorbidity: A Pre-Post Design. Int J Environ Res Public Health, 2022. 19(23).
- Hooker, S.A., SF-36, in Encyclopedia of Behavioral Medicine, M.D. Gellman and J.R. Turner, Editors. 2013, Springer New York: New York, NY. p. 1784-1786.
- Yoo, I., Social Networks as A Predictive Factor in Preserving Cognitive Functioning During Aging: A Systematic Review. Home Health Care Management & Practice, 2022. 34(1): p. 72-80.
- Schierz, O., K. Baba, and K. Fueki, Functional oral health‐related quality of life impact: A systematic review in populations with tooth loss. Journal of Oral Rehabilitation, 2021. 48(3): p. 256-270.
- Arranz, L.-I., M. Rafecas, and C. Alegre, Effects of Obesity on Function and Quality of Life in Chronic Pain Conditions. Current Rheumatology Reports, 2013. 16(1): p. 390.
- Chin, S.-H., et al., Obesity and pain: a systematic review. International Journal of Obesity, 2020. 44(5): p. 969-979.
- Sorel, J.C., et al., The influence of preoperative psychological distress on pain and function after total knee arthroplasty. The Bone & Joint Journal, 2019. 101-B(1): p. 7-14.
- Cohen, R.A., M.M. Marsiske, and G.E. Smith, Neuropsychology of aging. Handb Clin Neurol, 2019. 167: p. 149-180.
- Gariballa, S. and A. Alessa, Associations between low muscle mass, blood-borne nutritional status and mental health in older patients. BMC Nutrition, 2020. 6(1): p. 6.
- Roberts, T., et al., Factors associated with health service utilisation for common mental disorders: a systematic review. BMC Psychiatry, 2018. 18(1): p. 262.
- Nosraty, L., et al., Perceptions by the oldest old of successful aging, Vitality 90+ Study. Journal of aging studies, 2015. 32: p. 50-58.
- Haraldstad, K., et al., A systematic review of quality of life research in medicine and health sciences. Quality of Life Research, 2019. 28(10): p. 2641-2650.
- Theis, S., et al., Quality of life in menopausal women in the workplace - a systematic review. Climacteric, 2023. 26(2): p. 80-87.
We hope that we have addressed the first reviewers' comments in an appropriate manner, and we are looking forward to receiving your final decision on our manuscript.
Thank you very much and sincerely,
Petra Stute, M.D., Susanne Theis, M.D.
- Haraldstad, K., et al., A systematic review of quality of life research in medicine and health sciences. Quality of Life Research, 2019. 28(10): p. 2641-2650.

Reviewer 2 Report
Comments and Suggestions for Authors
Introduction
What was the objective of the study? It is not clear.
"Therefore, combined data from the Bern Cohort Study 2014 (BeCS-14) was used to generate a score from selected BFS items to represent QoL objectively." This is method.
Results
The word aging appears in the title, but the average age of the study was 39 years old. This is strange and means that the majority of participants were under 60.
Some results are repeated in the text and tables, and some appear different in the text compared to the tables.
Could the fact that the study included 73 women have affected the results? This was not discussed.
Conclusions
There are many sentences that are not a conclusion, but should be included in the discussion.
As the objectives are not clear, it is difficult to say whether the conclusion responds to them.
References
Of all references, around 64% have been published for 5 years or more. They need to review.
Author Response
We thank the reviewers for their valuable comments and suggestions for improvement. We have revised the manuscript according to the reviewers' suggestion. All changes have been highlighted in the manuscript. Enclosed are our responses point by point to the reviewers' comments and suggestions:
Reviewer 2
Introduction
Comment 1: What was the objective of the study? It is not clear.
Author´s reply:
We absolutely agree with the reviewer that a precise description of the objective is appropriate.
Quality of life is becoming increasingly important in medical research (e.g. cancer and cancer survivors, as well as side effects of treatments affecting QoL for example). To our knowledge, existing measurement instruments concerning QoL are patient reported outcome measures and therefore subject to subjective assessment by patients. For this reason, we wanted to establish an independent way of measuring quality of life apart from subjective evaluation to be able to use this in future research work.
We have therefore added the following to the Introduction (page 2 paragraph 1 line 59-61): For this reason, we wanted to establish an independent way of measuring quality of life apart from subjective evaluation by a patient. At best, this could be used for future research on objective QoL.
…
Page 2 paragraph 2 line 75 – 79 As previous research on quality of life and in particular on health-related quality of life has so far mainly used subjective methods, but objectifiable comparisons are equally important for scientific comparability we wanted to examine whether it is even possible to depict quality of life (which is per se subjective) with the help of objectifiable parameters.
And to Results Page 6 paragraph 3 line 219-221 (3.4): As described above (section 2.4) we aimed to statistically identify BFS item combinations that best represented each SF-36 subscale to enable the most objective measurement of hrQoL possible.
Comment 2: "Therefore, combined data from the Bern Cohort Study 2014 (BeCS-14) was used to generate a score from selected BFS items to represent QoL objectively." This is method.
Author´s reply:
We thank the reviewer for this valuable advice and adjusted as required: Therefore, combined data from the Bern Cohort Study 2014 (BeCS-14) was used to generate a score from selected BFS items to represent QoL objectively. (page 2 paragraph 2 line 73-75)
Results
Comment 3: The word aging appears in the title, but the average age of the study was 39 years old. This is strange and means that the majority of participants were under 60.
Author´s reply:
Thanks for this very important hint. We fully agree with the reviewer about this confusing designation. Thus, we decided to change the title of the manuscript. This also makes the purpose of the study clearer from the outset (see Comment 1).
We changed title as follows: Measuring quality of life incorporating objectively measurable parameters within the cross-sectional Bern cohort study 2014 (BeCS-14)
Page 1, headline, line 2-3
Comment 4: Some results are repeated in the text and tables, and some appear different in the text compared to the tables.
Author´s reply:
We appreciate the attention of the reviewer and are pleased that we can explain this as follows:
In the total cohort, missing categorical values were set as "unknown" and metric values were not replaced. Table 1a shows the values that represent the percentage of respondents affected by the respective statement. To avoid confusion for the reader, we have adapted the corresponding information in the text to the data presented in Table 1a.
“2… Briefly, the majority of the total cohort was female (73.7%), mean age was 39.5 (SD 14.9 %) years, 55.6% were married or lived in a permanent partnership, 64,9% were childless, more than 50% were well educated and had a monthly gross income above 5000 CHF. More than 50% worked in the social or economy sector with 51,7% being employed. Job occupation was at least 90% in 41.6% and at least 50% in 26,7% of participants. The majority was non-smoking (65.0%) and physically active (74,6%). 35,2% consumed alcohol daily. …”
(Page 4 Paragraph 4 Line 176 – 181)
Table 3 showed the values at an arbitrary point in time of the optimization, which we have now supplemented so that it contains the values after optimization and standardization as in the text. Table 3 shows that it is sufficient to perform the analyses for TC (because the correlation coefficients of SG1 and SG2 are similar).
(Page 7 Paragraph 2 Line 233-234)
Comment 5: Could the fact that the study included 73 women have affected the results? This was not discussed.
Author´s reply: In answering this question, we assume that it is meant whether a proportion of 74 percent women in the total cohort (73 % in S1 and 80 % in S2) could have influenced the results.
The reviewer raises an important point. The unequal distribution of men and women could of course influence the results. Therefore, we tested the influence of gender and age for each subscale of the SF-36. All regression models proved to be statistically significant so the models, differentiated by gender, are comparable. There was no influence of sex except in one subscale (bodily pain). This might be due to the fact that regarding this subscale more subjective than objective parameters were analysed.
We also mentioned this in the Discussion page 13 paragraph 3 line 496: “… Age and sex have no material impact on correlation coefficients in this investigation (except the two subscales PF and BP of SF-36, age had an impact on correlation in males as well as sex had an influence in subscale BP of SF-36). Therefore, we postulate that our findings are a stable and objective option to measure QoL. Limited applicability for measuring quality of life in men might apply for the two subscales PF and BP. On the other hand, the proportion of men in the overall cohort is lower, so this could also be a cause for this result. …”
Conclusions
Comment 6: There are many sentences that are not a conclusion, but should be included in the discussion.
Author´s reply:
We thank the reviewer for this important hint and have adjusted conclusions as follows (page 13 paragraph 4 line 515-525): Health-related QoL is usually assessed by (validated) questionnaires thus only incorporating an individual’s subjective ratings. Fortunately, as quality of life moves to the forefront of clinical research, its objectifiability and, above all, objective measurability become important for future research projects.
Based on the cross-sectional cohort study BeCS-14 described above and the validated questionnaire SF-36, As described above we demonstrated that hr-QoL could also be assessed by objective parameters or a combination of subjective and objective parameters. To our knowledge we are the first to correlate a validated set of 34 objective and 9 subjective parameters with subjectively evaluated SF-36 subscales. Out of these, 21 BFS items (13 objective and 8 subjective BFS items) were identified to significantly correlate at least once with the SF-36 subscales. The objectifiability of the SF-36 questionnaire demonstrate that questions on quality of life can be answered independently of a subjective assessment by subjects in future scientific studies. The deleted sentences have been integrated into the discussion or were already integrated for the most part.
Comment 7: As the objectives are not clear, it is difficult to say whether the conclusion responds to them.
Author´s reply:
After having specified the objectives as described in comment 1, we believe that the also adjusted conclusions as described in comment 6 now appear more logical and fitting.
References
Comment 8:
Of all references, around 64% have been published for 5 years or more. They need to review.
Author´s reply:
We thank the reviewer for this valuable hint and agree with the fact that this is striking. After thorough examination and further re-reviewing the literature, we can justify this as follows:
Both, the validation studies of SF-36 as well as the establishment and content of BFS have been performed in the years before 2017 and therefor refer to references before that year. This fact also justifies the necessary future review and optimization of research into quality of life and, in particular, its measurability. Thus, this fact ultimately supports our opinion that other, standardized and at best more objective methods should be established to measure quality of life in particular in health-related quality of life. This fact is also supported by other recent publications [1].
In total 59,3 % of the references refer to 2017 or the years before. To view the literature that does not describe the establishment, validation or domains of the two mentioned measuring methods (coloured in light blue, see below), 26 of the cited references remain. Thereof 8 references (30,7 %) are of the year 2017 or before. Of the 26 cited references 2 are older than ten years, which means that 92,3% of the used references are published within the last ten years. We think that these facts definitely support the need for further renewing of validated tools in research of quality of life.
References
- Felce, D. and J. Perry, Quality of life: Its definition and measurement. Research in Developmental Disabilities, 1995. 16(1): p. 51-74.
- Kuh, D., et al., A life-course approach to healthy ageing: maintaining physical capability. Proc Nutr Soc, 2014. 73(2): p. 237-48.
- Stute, P., et al., Measuring Active and Healthy Ageing: Applying a generic interdisciplinary assessment model incorporating ICF. The journal of nutrition, health & aging, 2017. 21(9): p. 1002-1009.
- Stute, P., et al., Measuring chronic stress exposure incorporating the active and healthy ageing (AHA) concept within the cross-sectional Bern cohort study 2014 (BeCS-14). BioPsychoSocial Medicine, 2019. 13(1): p. 2.
- Stute, P., et al., Measuring cognitive performance in way that incorporates the concept of active and healthy ageing (AHA). Maturitas, 2019. 125: p. 27-32.
- Arifi, D., et al., Impact of chronic stress exposure on cognitive performance incorporating the active and healthy aging (AHA) concept within the cross-sectional Bern Cohort Study 2014 (BeCS-14). Archives of Gynecology and Obstetrics, 2022. 305(4): p. 1021-1032.
- Mathieu, L., et al., Illness perception in overweight and obesity and impact on bio-functional age. Maturitas, 2017. 103: p. 92-93.
- Hamilton, M., A RATING SCALE FOR DEPRESSION. Journal of Neurology, Neurosurgery & Psychiatry, 1960. 23(1): p. 56.
- Bullinger, M., I. Kirchberger, and J. Ware, Der deutsche SF-36 Health Survey Übersetzung und psychometrische Testung eines krankheitsübergreifenden Instruments zur Erfassung der gesundheitsbezogenen Lebensqualität. Zeitschrift für Gesundheitswissenschaften= Journal of public health, 1995. 3(1): p. 21-36.
- Schulz, P., W. Schlotz, and P. Becker, Trierer Inventar zum chronischen stress (TICS)[Trier inventory for chronic stress (TICS)]. 2004.
- Ardelt-Gattinger, E., et al., AD–EVA-KJ: Interdisziplinäres Testsystem zur Diagnostik und Evaluation bei Adipositas und anderen durch Ess-und Bewegungsverhalten beieinflussbaren Krankheiten im Kindes-und Jugendalter. Monatsschrift für Kinderheilkunde, 2009. 157(Suppl. 2): p. 206.
- Zenz, H., C. Bischoff, and V. Hrabal, Patiententheoriefragebogen:(PATEF). 1996: Hogrefe, Verlag für Psychologie.
- Molnar, M., C. Haiden, and B. Geißler-Gruber, IMPULS-Broschüre und IMPULS-Test. Betriebliche Analyse der Arbeitsbedingungen. 2012, AUVA, AK, ÖGB, WKÖ (Hg.) Wien.
- Camm, A.J., et al., Heart rate variability: standards of measurement, physiological interpretation and clinical use. Task Force of the European Society of Cardiology and the North American Society of Pacing and Electrophysiology. 1996.
- Kalbe, E., et al., Das Inventar zur Gedächtnisdiagnostik: Vorstellung der endgültigen Version. Aktuelle Neurologie, 2006. 33(S 1): p. P301.
- Morfeld, M. and M. Bullinger, Der SF36 Health Survey zur Erhebung und Dokumentation gesundheitsbezogener Lebensqualität. Physikalische Medizin Rehabilitationsmedizin Kurortmedizin - PHYSIK MED REHABIL KURORTMEDI, 2008. 18: p. 250-255.
- Ware Jr, J.E. and C.D. Sherbourne, The MOS 36-item short-form health survey (SF-36): I. Conceptual framework and item selection. Medical care, 1992: p. 473-483.
- O'Dea, I., M.S. Hunter, and S. Anjos, Life satisfaction and health-related quality of life (SF-36) of middle-aged men and women. Climacteric, 1999. 2(2): p. 131-40.
- Poethig, D., Experimentelle Entwicklung eines klinischen Diagnostikmodelles zur Objektivierung des biologischen Alters des Menschen : (e. nicht-invasives Komplexverfahren zur ambulanten gerontolog. Funktionsdiagnostik). 1984, Leipzig, Univ., Diss. B, 1984 (Nicht f.d. Austausch).
- Ries, W. and D. Pöthig, Chronological and biological age. Exp Gerontol, 1984. 19(3): p. 211-6.
- Dean, W., et al., Biological aging measurement: Clinical applications. 1988: Center for Bio-Gerontology.
- Meißner-Pöthig, D., Vitalität und ärztliche Intervention: Vitalitätsdiagnostik: Grundlagen, Angebote, Konsequenzen. 1996: Hippokrates-Verlag.
- Ware, J.E.J., et al. How to score and interpret single-item health status measures: a manual for users of the SF-8™ Health Survey. 2001.
- Ware, J., et al., SF36 Health Survey: Manual and Interpretation Guide. Lincoln, RI: Quality Metric, Inc, 1993, 1993. 30.
- Simon, G.E., et al., SF-36 Summary Scores: Are Physical and Mental Health Truly Distinct. Medical Care, 1998. 36(4): p. 567-572.
- Wilson, D., J. Parsons, and G. Tucker, The SF-36 summary scales: Problems and solutions. Sozial- und Präventivmedizin, 2000. 45(6): p. 239-246.
- Fluckiger, L., et al., Differential Effects of Aging on Heart Rate Variability and Blood Pressure Variability. The Journals of Gerontology: Series A, 1999. 54(5): p. B219-B224.
- Marchais, S.J., et al., Arterial Compliance and Blood Pressure. Drugs, 1993. 46(2): p. 82-87.
- Mariano, I., et al., BLOOD PRESSURE RESPONSES TO STRESS AFTER CHRONIC PHYSICAL EXERCISE: A SYSTEMATIC REVIEW WITH META-ANALYSIS. Journal of Hypertension, 2021. 39: p. e402.
- Poethig, D., Experimental development of a clinical diagnostic model objectifying bio-functional age (ing) of human being. 1984, Habilitation thesis (dt.). German: National Library Leipzig.
- Roca, J.A.R., et al., Oral Status of Elderly Patients In Long Stay Centers: A Systematic Review. 2020.
- Peres, M.A., et al., Oral diseases: a global public health challenge. The Lancet, 2019. 394(10194): p. 249-260.
- Tan, H., K.G. Peres, and M.A. Peres, Retention of Teeth and Oral Health–Related Quality of Life. Journal of Dental Research, 2016. 95(12): p. 1350-1357.
- Owsley, C., Vision and Aging. Annual Review of Vision Science, 2016. 2(1): p. 255-271.
- Swenor, B.K., et al., Aging With Vision Loss: A Framework for Assessing the Impact of Visual Impairment on Older Adults. The Gerontologist, 2019. 60(6): p. 989-995.
- Antonovsky, A., Unraveling the mystery of health: How people manage stress and stay well. 1987: Jossey-bass.
- Schumacher, J., et al., The Antonovsky Sense of Coherence Scale. Test statistical evaluation of a representative population sample and construction of a brief scale. Psychotherapie, Psychosomatik, Medizinische Psychologie, 2000. 50(12): p. 472-482.
- Singer, S. and E. Brähler, Die» Sense of Coherence Scale «: Testhandbuch zur deutschen Version. 2007: Vandenhoeck & Ruprecht.
- Blüher, M., Obesity: global epidemiology and pathogenesis. Nature Reviews Endocrinology, 2019. 15(5): p. 288-298.
- Mancuso, P. and B. Bouchard, The impact of aging on adipose function and adipokine synthesis. Frontiers in endocrinology, 2019. 10: p. 137.
- Ponti, F., et al., Aging and imaging assessment of body composition: from fat to facts. Frontiers in endocrinology, 2020. 10: p. 861.
- Pizarro-Mena, R., et al., Effects of a Structured Multicomponent Physical Exercise Intervention on Quality of Life and Biopsychosocial Health among Chilean Older Adults from the Community with Controlled Multimorbidity: A Pre-Post Design. Int J Environ Res Public Health, 2022. 19(23).
- Hooker, S.A., SF-36, in Encyclopedia of Behavioral Medicine, M.D. Gellman and J.R. Turner, Editors. 2013, Springer New York: New York, NY. p. 1784-1786.
- Yoo, I., Social Networks as A Predictive Factor in Preserving Cognitive Functioning During Aging: A Systematic Review. Home Health Care Management & Practice, 2022. 34(1): p. 72-80.
- Schierz, O., K. Baba, and K. Fueki, Functional oral health‐related quality of life impact: A systematic review in populations with tooth loss. Journal of Oral Rehabilitation, 2021. 48(3): p. 256-270.
- Arranz, L.-I., M. Rafecas, and C. Alegre, Effects of Obesity on Function and Quality of Life in Chronic Pain Conditions. Current Rheumatology Reports, 2013. 16(1): p. 390.
- Chin, S.-H., et al., Obesity and pain: a systematic review. International Journal of Obesity, 2020. 44(5): p. 969-979.
- Sorel, J.C., et al., The influence of preoperative psychological distress on pain and function after total knee arthroplasty. The Bone & Joint Journal, 2019. 101-B(1): p. 7-14.
- Cohen, R.A., M.M. Marsiske, and G.E. Smith, Neuropsychology of aging. Handb Clin Neurol, 2019. 167: p. 149-180.
- Gariballa, S. and A. Alessa, Associations between low muscle mass, blood-borne nutritional status and mental health in older patients. BMC Nutrition, 2020. 6(1): p. 6.
- Roberts, T., et al., Factors associated with health service utilisation for common mental disorders: a systematic review. BMC Psychiatry, 2018. 18(1): p. 262.
- Nosraty, L., et al., Perceptions by the oldest old of successful aging, Vitality 90+ Study. Journal of aging studies, 2015. 32: p. 50-58.
- Haraldstad, K., et al., A systematic review of quality of life research in medicine and health sciences. Quality of Life Research, 2019. 28(10): p. 2641-2650.
- Theis, S., et al., Quality of life in menopausal women in the workplace - a systematic review. Climacteric, 2023. 26(2): p. 80-87.
Authors response to Questions for General Evaluation
|
2. Questions for General Evaluation |
Reviewer’s Evaluation |
Response and Revisions |
|
Does the introduction provide sufficient background and include all relevant references? |
Must be improved |
Please see authors’ answer Reviewer 2 Comment 1 and the resulting changes highlighted in the manuscript |
|
Are all the cited references relevant to the research? |
Must be improved |
Please see authors’ answer Reviewer 2 Comment 8 |
|
Is the research design appropriate? |
Must be improved |
Please see authors’ answer Reviewer 1 Comment 5 |
|
Are the methods adequately described? |
Must be improved |
Please see highlighted changes in manuscript 1.0 Introduction/ 2.4 Statistical methos / 3.4 Incorporating SF-36… |
|
Are the results clearly presented? |
Yes/Can be improved/Must be improved/Not applicable |
- |
|
Are the conclusions supported by the results? |
Yes/Can be improved/Must be improved/Not applicable |
- |
We hope that we have addressed the first reviewers' comments in an appropriate manner, and we are looking forward to receiving your final decision on our manuscript.
Thank you very much and sincerely,
Petra Stute, M.D., Susanne Theis, M.D.
- Haraldstad, K., et al., A systematic review of quality of life research in medicine and health sciences. Quality of Life Research, 2019. 28(10): p. 2641-2650.

Round 2
Reviewer 1 Report
Comments and Suggestions for Authors
1. Sentence is unclear: Job occupation was at least 90% in 41.6% and at least 50% in 26,7% of participants.
2. It is possible to test and indicate significant differences in % between subgroup 1 and subgroup 2 in Table 1a. And then discuss how differences between the subgroups may impact results.
